# Hydrological and topographic determinants of biomass and species richness in a Mediterranean-climate shrubland

Samantha Díaz de León-Guerrero[1], Rodrigo Méndez-Alonzo [1]*, Stephen H. Bullock[1], Enrique R. Vivoni [2]

1 Departamento de Biología de la Conservación, Centro de Investigación Científica y de Educación Superior de Ensenada (CICESE), Ensenada, Baja California., México, 2 School of Earth and Space Exploration & School of Sustainable Engineering and the Built Environment, Arizona State University, Tempe, Arizona, United States of America

* mendezal@cicese.mx

**Data Availability Statement:** All relevant data are within the paper and its Supporting information files.

## Abstract

### Background

In arid and semiarid shrublands, water availability directly influences ecosystem properties. However, few empirical tests have determined the association between particular soil and hydrology traits with biodiversity and ecosystem biomass at the local scale.

### Methods

We tested if plant species richness ($S$) and aboveground biomass (AGB) were associated with soil and topographic properties on 36 plots (ca. 12.5 m$^2$) in 17 hectares of chaparral in the Mediterranean-climate of Valle de Guadalupe, Baja California, México. We used close-to-the-ground aerial photography to quantify sky-view cover per species, including all growth forms. We derived an elevation model (5 cm) from other aerial imagery. We estimated six soil properties (soil water potential, organic matter content, water content, pH, total dissolved solids concentration, and texture) and four landscape metrics (slope, aspect, elevation, and topographic index) for the 36 plots. We quantified the biomass of stems, leaves, and reproductive structures, per species.

### Results

86% of AGB was in stems, while non-woody species represented 0.7% of AGB but comprised 38% of $S$ (29 species). Aboveground biomass and species richness were unrelated across the landscape. $S$ was correlated with aspect and elevation ($R = 0.53$, aspect $P = 0.035$, elevation $P = 0.05$), while AGB (0.006–9.17 Kg m$^{-2}$) increased with soil water potential and clay content ($R = 0.51$, $P = 0.02$, and $P = 0.04$). Only three species (11% of total $S$) occupied 65% of the total plant cover, and the remaining 26 represented only 35%. Cover was negatively correlated with $S$ ($R = -0.38$, $P = 0.02$). 75% of AGB was concentrated in 30% of the 36 plots, and 96% of AGB corresponded to only 20% of 29 species.

**Funding:** RMA 278755 Fondo Sectorial CONACYT INEGI Website: https://www.inegi.org.mx/investigacion/conacyt/default.html SDDLG 274874 CONACYT Scholarship for PhD students NO: The funders had no role in study design, data collection and analysis, decision to publish, or preparation of the manuscript.

**Competing interests:** The authors have declared that no competing interests exist.

## Discussion

At the scale of small plots in our studied Mediterranean-climate shrubland in Baja California, AGB was most affected by soil water storage. AGB and cover were dominated by a few species, and only cover was negatively related to $S$. $S$ was comprised mostly by uncommon species and tended to increase as plant cover decreased.

## Introduction

Water availability is linked to critical ecosystem properties and functions [1]. Exploring how plant species richness ($S$) and aboveground biomass (AGB) are associated with water-related substrate properties is critical to quantifying the potential of ecosystems to store carbon and their vulnerability to anthropogenic stressors [2–4]. At the continental and regional scales, vegetation biomass and species diversity are correlated and co-vary with climate variables, such as solar irradiance and precipitation [5–7]. At local scales, two hypotheses have been proposed to explain the biodiversity and ecosystem function relationship: a) the mass-ratio hypothesis [8] indicates that resource dynamics is a function of the structural and physiological traits of the dominant set of species, such that ecosystem properties, including biomass, depend on keystone species; and, b) the niche complementary hypothesis [9] indicates that resources are progressively partitioned across species, such that the highest $S$ maximizes ecosystem functions. Other factors contributing to $S$ are elevation [10], landscape heterogeneity, field management, habitat composition [11], competition between species, and different disturbance levels [12].

Across global drylands, the aboveground net productivity variability is a linear function of precipitation [13], and $S$ determines several ecosystem functions, such as nutrient cycling and carbon storage, yet the relationship between biodiversity to productivity remains unclear [14]. At the regional scale in Mediterranean-climate drylands, $S$ varies in response to the abiotic conditions and microclimates due to topography and soil characteristics [15, 16]. Similarly, AGB in different dryland environments of Europe and the Americas is positively related to greater biodiversity [17], while regional studies have found a positive relationship with annual precipitation [18] and with soil water storage over 0 to 100 cm depths [19]. However, at the plot scale, evidence demonstrates that $S$ and AGB are unrelated [20], suggesting that different environmental drivers operate locally to promote these two ecosystem properties.

In drylands, hydrological networks determine the spatial distribution of vegetation and the landscape ecology of arid vegetation mosaics [21]. In such conditions, higher species diversity and biomass may be promoted by positive-reinforcement processes, such as tree establishment reducing bare soil evaporation, thus helping maintain soil moisture which increases the probability of plant establishment, particularly in locations where water accumulates or where the phreatic level is close to the surface [22–25]. In addition, vegetation patches typically have higher infiltration capacity than bare soils and deep-rooted plants can increase shallow soil moisture by redistributing water from deeper soil layers by an ecophysiological mechanism termed hydraulic lift [26, 27]. These synergistic processes may promote soil heterogeneity and reinforce vegetation growth, analogous to the formation of islands of fertility dispersed across the landscape [28]. This type of facilitative interactions caused by species diversity of functional groups are relevant to the understanding of the long-term population dynamics and the accumulation of biomass within drylands [29–31].

There are still gaps in our understanding of the relationship between *S* and productivity or accumulated biomass in semiarid environments. Few studies have explored how *S* and AGB of drylands are associated with hydrologic and soil properties within field settings [3, 4]. Greenness [32] and productivity depend in part on soil moisture [33], which is affected by topography, soil, and geology [34]. The relation of *S* to local topographic variation is scarcely known [35, 36], as previous work has focused on associations and species across regional settings [37]. However, with the advent of new remote sensing technologies and geostatistical methods, it is feasible to test how *S*, AGB, soil properties, and terrain attributes co-vary across small distances [38]. In particular, high-resolution remote sensing obtained from unmanned aerial vehicles (UAVs) and near ground digital cameras may allow generating spatial distributions of *S*, AGB, and digital elevation models [39–41]. In conjunction with geospatial techniques [42], these tools are useful to explore the set of factors that may influence the abundance and distribution patterns of plant species in shrublands.

In this study, we quantified the relationships between *S*, AGB, soil, and terrain variables in 36 plots within a ca. 17-hectare site in a chaparral of the Valle de Guadalupe in Baja California, México, a region with high variability in rainfall, characterized by a long dry season from April to November, and a rainy period between December and March. The chaparral, a type of semi-arid shrubland of California and Baja California mainly composed of evergreen sclerophyllous, short-stature perennial plants, varies in physiognomy and diversity due to precipitation and radiation [43]. It has recently been negatively affected by an extraordinary drought spanning over ten years [44, 45]. However, covariation between AGB and *S* in chaparral is not fully understood, nor are the patterns of association between AGB-*S* with abiotic factors, such as water availability, soil texture, or topographic indices. Our primary hypothesis follows the niche complementarity hypothesis, as we expect that both *S* and AGB would increase with higher water availability: plots with higher water content would sustain more species and would also support a larger AGB [3, 7, 46, 47] due to the complementarity of functional groups. If this latter hypothesis is supported, we would expect *S* and AGB to increase with higher soil water and nutrient availability, quantified by the soil water potential, organic matter content, soil water content, pH, total dissolved solids concentration, and percent of sand, clay and silt, and *S* and AGB would also increase in sites protected from solar radiation, and where water accumulates (determined by the slope, aspect, elevation, and topographic index). In conjunction, our results would allow the identification of the putative abiotic determinants of *S* and AGB in chaparrals and other drylands.

## Materials & methods

### Study site

The study was conducted in private property, with express permission from the owner. Owner is acknowledged in the text (Mrs. Natalia Badan, Valle de Guadalupe, Mexico). No endangered species were collected in this study. Our study site is a 17 ha plot in chaparral within Rancho El Mogor (32˚ 1'49.95" N, 116˚36'16.56" W) in the Valle de Guadalupe, Baja California, México, 16.5 km from the Pacific coast (Fig 1A). The Valle de Guadalupe has a semiarid Mediterranean climate with warm, dry summers and cool, moist winters [48] and Csa Köppen-type climate (hot-dry summers Mediterranean-type climate [49]). The mean annual temperature has been 17.9 ˚C (ranging from 12.2 to 24.9 ˚C), and the mean annual precipitation was 298 mm (1986–2016 from El Porvenir and Agua Caliente meteorological stations, ca. 15 km distant), with most rainfall occurring from December through March [32]. Mean monthly precipitations in winter were 18–63 mm (from November thru April), and 1–6 mm in the dry summers (May thru October) for the period of 1980–2009 [50]. We located our study site on the transition

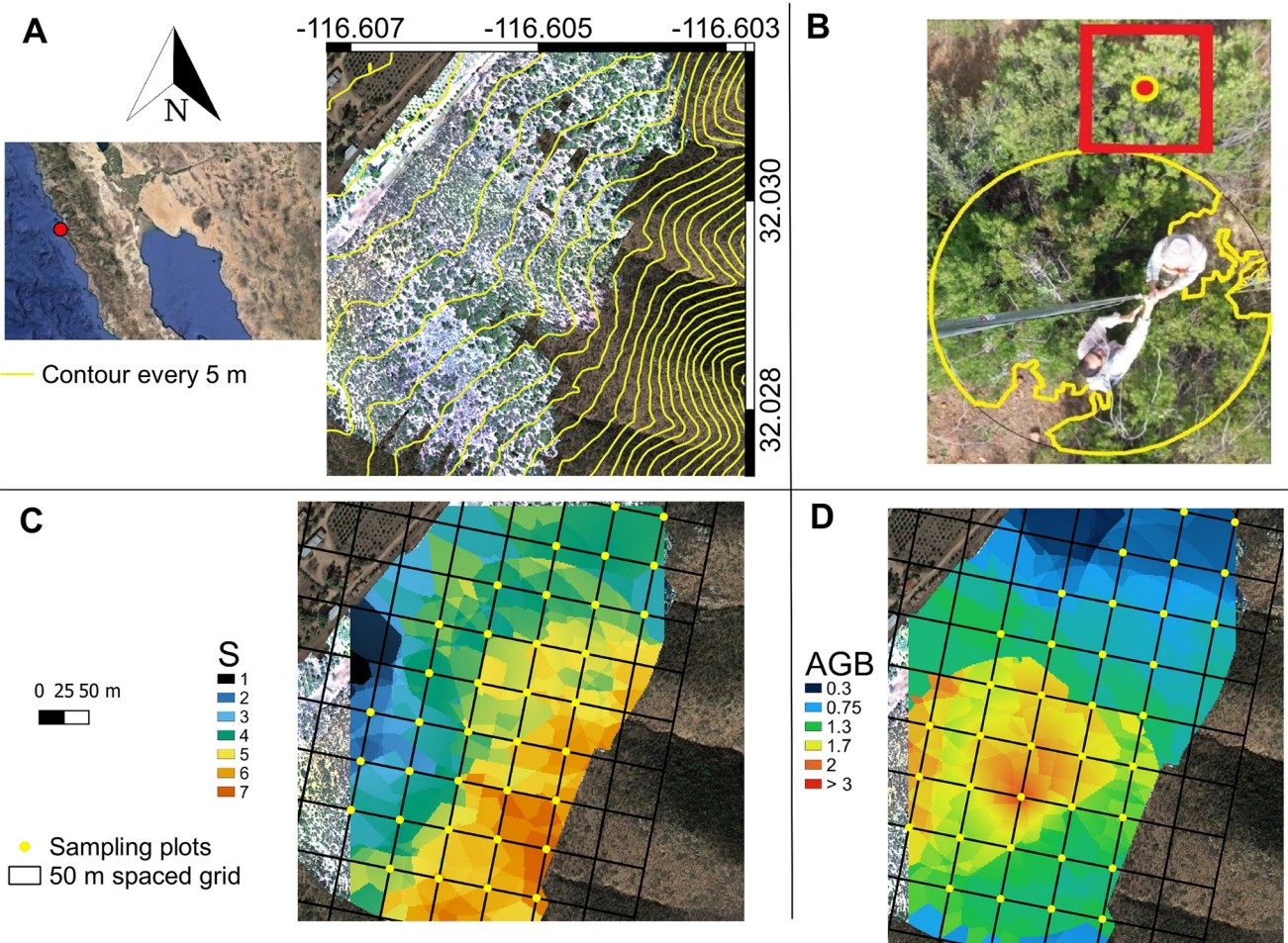

**Fig 1. Location of the study site and distribution of 36 sampling plots at Rancho El Mogor, Baja California, México.** A: 36 plots were on the intersections of a 50 m rectangular grid; B: experimental design to quantify species richness (*S*) and vegetation cover in our plots (2 m radius, ca. 12.5 m$^2$). Aboveground biomass (AGB) was harvested in one square meter located at the North edge of the sampling plots. At the center of the square, one soil sample was extracted to 25 cm depth. Interpolation maps of C: *S*, and D: AGB (kg m$^{-2}$) based on Empirical Bayesian Kriging. The individual pictured in Fig 1 has provided written informed consent (as outlined in PLOS consent form) to publish their image alongside the manuscript.

from steep hilly terrain on granitic rocks to the valley floor, with a generally West aspect and inclination of 7.5˚. The site was at the sharp border between large expanses of native shrubland composed of chaparral and coastal scrub [51] and agricultural land use. It was last burned in 1988 and has been traversed or browsed a few days of the year by a small herd of cattle inclined to forage in surrounding areas that are more verdant or tended. The site approximates the core of the footprint area of an eddy covariance CO2 flux study [52] that has micrometeorological records including soil water content (but lacking replicates to compare with our samples).

### Initial mapping

To delimit our study site at the beginning of the experiment, we produced a digital elevation model (DEM) with 5 cm horizontal and vertical resolution, derived from 108 geotagged photographs taken at 40 m height (Sony EXMOR RGB 12.4-megapixel camera, 20 mm 94˚ FOV lens) from a UAV (Phantom 3, FC300S, DJI, Shenzhen, China), during a single flight in

October, 2016. Photographs were mosaicked and ortho-corrected with the coordinates from 14 checkpoints obtained from a GPS survey (Pathfinder model ProXH, Trimble, Sunnyvale, CA, USA). The DEM was produced using the Lastools software [53]. Contour lines were obtained with ArcGIS [54].

Additionally, we used a DEM at 5 m resolution from the National Institute of Statistics, Geography and Informatics, México [55]. With the use of QGIS 3.12 [56], we extracted the elevation, aspect, contour lines, slope, and a topographic index, which is a relative measure of moisture at a pixel where water accumulation is due to the upslope contributing hydrologic network and the slope of the pixel itself [57, 58]. The plots (N = 36) for species presence and cover, and adjacent AGB and soil sampling, were set on the intersections (at 50 m) of a rectangular grid to facilitate geospatial analysis (Fig 1A). The sampling plots were all within the chaparral area of the ranch. The elevation ranged from 395.6 to 429 m.a.s.l., with the aspect inclined to face 166˚ to 301˚, and slope ranged from 4˚ to 26˚. The topographic index showed values from -0.39 to 0.21 (Table 1, S1 Table).

## Species presence and cover

$S$ for each plot was determined by direct observation (April 2017), including all growth forms. In our study site, November to April is the time of the year when greenness rises and peaks [32, 50]. During the same sampling, species cover was quantified using close-to-the-ground vertical images, taken at 5 m aboveground, centered over the plot, with a 5-megapixel camera (J5 Smartphone, Samsung Electronics, Seoul, Korea). Within an area of 2 m radius (ca. 12.5 m$^2$) per each plot, species cover was roughly delimited in the field and later refined using ImageJ ([59] Fig 1B). The individual pictured in Fig 1 has provided written informed consent (as outlined in PLOS consent form) to publish their image alongside the manuscript. This photographic procedure, similar to intra-site photography in archaeology [60] allowed us to calculate the shrub canopy area more precisely than the conventional calculations of canopy area based on the quantification of major ratio vs. minor ratio [61, 62]. In addition, discrimination among species occupying the canopy was feasible due to the resolution and of colors of the

**Table 1. Variables used in the study.** Set of variables measured on 36 plots in 17 hectares of native shrubland in Rancho El Mogor, Baja California, México.

| Variable | Abbreviation | Units | Mean | Min | Max | Median | Standard Deviation |
|---|---|---|---|---|---|---|---|
| Richness | $S$ | Number of species | 3.2 | 1 | 7 | 3.22 | 1.45 |
| Aboveground biomass | AGB | kg m$^{-2}$ | 1.15 | 0.006 | 9.17 | 0.69 | 1.69 |
| Cover | Cover | % per plot | 62.43 | 19.71 | 93.6 | 0.63 | 18.4 |
| Height | H | M | 1.47 | 0.89 | 2.42 | 1.35 | 0.42 |
| Soil water potential | $\Psi_{soil}$ | MPa | -37.94 | -71.47 | -5.66 | -32.26 | 19.47 |
| Soil organic matter | OM | % | 5.54 | 2.2 | 15.2 | 4.93 | 2.73 |
| pH | pH | Log H$^+$ | 6.42 | 6.09 | 6.97 | 6.45 | 0.21 |
| Total Dissolved Solids | TDS | mS | 29.15 | 10.67 | 188.58 | 21.25 | 29.25 |
| Soil Water Content | SWC | % | 0.0002 | 0 | 0.0013 | 0.000085 | 0.003 |
| Leaf Area | LA | m$^2$ m$^{-2}$ | 0.76 | 0.02 | 4.92 | 0.57 | 0.88 |
| Sand | Sand | % | 74.87 | 67.1 | 88.2 | 74.1 | 4.93 |
| Clay | Clay | % | 9.77 | 6.54 | 13.18 | 9.9 | 1.64 |
| Silt | Silt | % | 15.35 | 1.62 | 22.04 | 16.16 | 4.14 |
| Elevation | Elev | m.a.s.l. | 408.8 | 395.6 | 429.2 | 407.7 | 8.75 |
| Aspect | Aspect | ° | 228.1 | 166.1 | 301.8 | 228.3 | 27.75 |
| Slope | Slope | ° | 8.9 | 4.08 | 26.08 | 7.85 | 4.03 |
| Topographic index | TopoIndex | Unitless | -0.04 | -0.39 | 0.21 | -0.03 | 0.13 |

images. We also measured individual plant heights in the area. The timing corresponded to the end of the wet season when the maximum number of species would be evident. The nomenclature of plant species follows Rebman, Gibson, and Rich [63].

## Aboveground biomass (AGB) harvesting

AGB was harvested manually during from one square meter located 2.2 m North of the center of each of the 36 plots (Fig 1B), during February 2018, corresponding to the period of peak biomass at our study site [32, 64]. All material was transported to the laboratory within the same day and was separated by plot and species into stems, leaves, litter, and reproductive structures (flowers, fruits, and floral buds). The majority of species had flowers and floral buds, but there was only one species that was already in the transition from flowers to fruit (*Cneoridium dumosum*). Litter was almost all easily classified to species as well as tissue (leaves, twigs or woody fragments). Fresh weight was obtained the same day of collection, and the material was oven-dried at 70 ˚C for three days to quantify the dry weight, using an electronic scale with a resolution of 0.1g (Polder KSC-348-95US, China). Leaf area was estimated from the dry leaf mass, considering the specific leaf area of each species harvested according to Pérez-Harguindeguy et al. [65]. Eight of the species encountered in the survey of S were not found in the AGB sampling, including herbaceous perennials (*Xanthisma junceum* and a vine *Marah macrocarpa*), one acaulescent rosette perennial (*Hesperoyucca whipplei*), and deciduous shrubs (*Romneya trichocalyx*, *Trichostema parishii*, *Bahiopsis laciniata*, and *Encelia californica*).

## Soil properties

Soil samples were extracted during August 2018, corresponding to the late dry season when water availability becomes most limiting for plants. In particular, 2018 was an abnormally dry year in Baja California, with total precipitation of 95 mm in the Valle de Guadalupe, and no rain for 109 days before the sampling (data from a CICESE station ca. 3.3 km distant, 32˚ 0'0.00 N, 116˚36'10.00" W). Samples were extracted to 25 cm depth inside the harvested square meter (Fig 1B) using a soil auger, avoiding rocks and visible organic matter accumulations. Soil samples were sealed in plastic bags and transported to the laboratory the same day. In the laboratory, soil water potential was determined using a dew-point potentiometer (WPC-4 Dew Point Potential, Meter Group, Pullman WA, USA). pH and TDS were measured in a soil solution of 1:3 soil: water weight ratio. pH was measured with Oakton pH 450 (Oakton Instruments, Vernon Hills, IL, USA). Total dissolved solids (TDS) conductivity in solution was measured with a conductivity meter (Starter 300C, Ohaus, Parsippany, NJ, USA). Soil water content was measured by the difference of fresh mass and dry mass divided by the dry mass, after 72 hours at 70 ˚C. Organic matter was measured by the difference of weight after ignition in a furnace at 300 ˚C for three hours (2–525, J.M. NEY furnace, Tucson, AZ, USA). Soil particle size distribution was obtained by the hydrometer method [66] following adaptations from USDA [67]. See Table 1 for measurements and their units.

## Statistical analyses

Species accumulation curves were employed to evaluate the sampling effort. S of the study site was calculated via sample-based rarefaction [68] and compared with three asymptotic estimators of total richness (Fig 2A): Incidence-Based Coverage Estimator (ICE), a non-parametric adjustment model based on first-order Jackknife (Jackknife 1) and an asymptotic adjustment model (Chao1). ICE is based on the number of rare species (those present in less than ten plots), Chao 1 is an estimator based on rare species abundance, and Jackknife 1 is a function of the number of species found in only one plot [69].

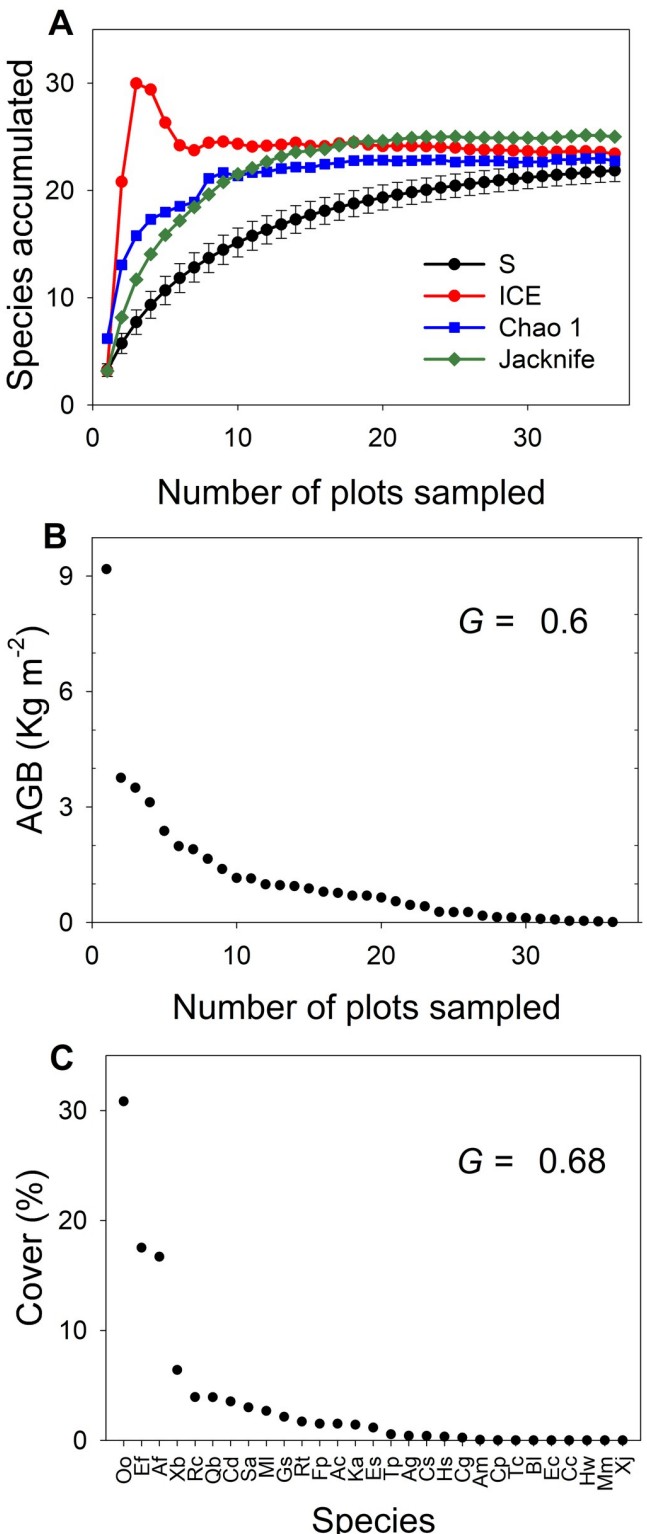

**Fig 2. Ecologic diversity in El Mogor, Valle de Guadalupe, Baja California, México.** A: Estimation of *S* of a chaparral community based on three asymptotic estimators: ICE, Chao 1, and Jackknife 1. B: Inequality of the distribution of AGB (kg m$^{-2}$) among the plots with plots ranked by AGB. C: Inequality of the distribution of relative cover (%) among the 29 species across the 36 plots. Species abbreviations in S2 Table.

*S* and AGB spatial distributions were displayed via interpolation with Empirical Bayesian Kriging using ArcGIS [54]. We used linear regression models to test for *S* and AGB relations and the relationship with soil and terrain properties, vegetation height, and plant cover with Pearson Correlation in JASP 0.8.6.0 [70]. To explore the set of abiotic variables associated with AGB and *S*, stepwise regression analyses were performed using the 'mass' [71] package (S1 Appendix) in R [72].

To quantify if the spatial distribution of *S* and AGB is concentrated in patches or homogeneously distributed, we calculated Gini coefficients (*G*) per each property. The Gini coefficient is an econometric indicator of data accumulation and skewness, where *G* = 1, indicates the highest inequality, implying that a single plot concentrated 100% of the property on the landscape [73]. In the opposite case, *G* = 0 would indicate that all measured plots have exactly the same amount of the property, e.g., denoting a perfectly uniform distribution of *S* or AGB. *G* for species relative cover, leaf area, and AGB per plot and per species were calculated using the 'ineq' (S2 Appendix) package [74] in R [72]. All graphical plots were generated using Sigma Plot 11.0 (Systat Software, San Jose, CA, USA).

## Results

### Variation in soil properties across the landscape

Soil water potentials ranged from -5.66 MPa in plots under *Quercus berberidifolia* to -71.47 MPa under various species, including *Ornithostaphylos oppositifolia*, *Eriodictyon sessilifolium*, and *Ceanothus greggii*. Soil organic matter ranged between 2.2 and 15.2%, and the samples were slightly acidic, with pH variation from 6.09 to 6.97. Total dissolved solids (TDS) varied between 10.67 and 188.58 mS. Sand content ranged from 67 to 88%, while clay and silt varied from 6 to 13 and 1 to 22%, respectively (Table 1, S1 Table).

We found no differences in soil texture classification among all samples collected (S1 Fig). Organic matter was positively related with soil water potential ($R = 0.47$, $P = 0.004$), TDS ($R = 0.45$, $P = 0.006$), soil water content ($R = 0.82$, $P < 0.001$; Table 2), but did not correlate with sand, clay or silt proportion, nor topographic variables. Also, soil water potential had positive relationship with soil water content ($R = 0.37$, $P = 0.02$) but did not correlate with the other soil and topographic variables. pH only co-varied negatively with the topographic index ($R = -0.39$, $P = 0.016$). TDS had a positive relation with soil water content ($R = 0.46$, $P = 0.004$; Table 2), but did not correlate with the other soil and topographic variables. Sand and silt were not related with the other soil and topographic variables, but clay did correlate with slope ($R = 0.37$, $P = 0.02$). Elevation had a positive relationship with slope ($R = 0.41$, $P = 0.01$), but not with the remaining topographic measurements (S3 Table).

### Species richness, cover, and local distribution

We found 29 species in the 36 plots (S2 Table), comprising 19 woody, nine herbaceous species, and one annual herbaceous plant species. Our estimation of the species richness (*S*), via the species accumulation curve and the asymptotic estimators of *S*, converged at ca. 30 species (Fig 2A). Individual plots had between one and seven species (Table 1). *Q. berberidifolia*, *Ceanothus greggii*, and *Acourtia microcephala* were found on only one plot, while *O. oppositifolia*, *Eriogonum fasciculatum*, *Adenostoma fasciculatum*, and *Gutierrezia sarothrae* occurred on 12 or more plots (frequency in S2 Table). According to the survey and the derived Empirical Bayesian Kriging interpolation map, plots with the highest *S* were mostly located in the highest elevations of the study site (Fig 1C). These plots contained a mixture of both woody and herbaceous species.

**Table 2. Species diversity and aboveground biomass relationships with biotic and abiotic variables.**

|  | $S$ | AGB | Stem mass | Cover | H | $\Psi_{soil}$ | OM | SWC | TDS | LA |
|---|---|---|---|---|---|---|---|---|---|---|
| $S$ | - | 0.11 | 0.1 | **-0.38** | -0.25 | -0.19 | 0.07 | -0.02 | 0.03 | 0.19 |
| AGB | | | **0.99**\*\* | 0.17 | **0.36** | **0.39** | **0.74**\*\* | **0.67**\*\* | 0.25 | **0.86**\*\* |
| Stem mass | | | | 0.18 | **0.37** | **0.39** | **0.74**\*\* | **0.67**\*\* | 0.24 | **0.84**\*\* |
| Cover | | | | | **0.43**\* | **0.37** | 0.33 | **0.36** | 0.13 | 0.08 |
| H | | | | | | **0.46**\* | **0.52**\* | **0.47**\* | 0.33 | 0.31 |
| $\Psi_{soil}$ | | | | | | | **0.47**\* | **0.37** | 0.05 | 0.33 |
| OM | | | | | | | | **0.82**\*\* | **0.45**\* | **0.72**\* |
| SWC | | | | | | | | | **0.46**\* | **0.62**\*\* |
| TDS | | | | | | | | | | 0.33 |

Patterns of correlation among species richness ($S$), aboveground biomass (AGB), stem dry mass, plant cover per plot (Cover), average shrub height (H), soil water potential ($\Psi_{soil}$), soil organic matter (OM), soil water content (SWC), soil conductivity due to total dissolved solids in solution (TDS), and leaf area (LA) on 36 plots in 17 hectares of native shrubland in Rancho El Mogor, Baja California, México. Bold: $P < 0.05$;

\* $P < 0.01$;

\*\*, P < 0.001.

Plant cover per plot (19–93%, Table 1) and $S$ were negatively associated ($R$ = -0.38, $P$ = 0.02; Table 2), and $S$ was not correlated to any other biotic variable. The plot with the highest plant cover (93%) had only one species (*Q. berberidifolia*), had the least negative soil water potential (-5.66 MPa), the highest soil water content (0.0013%) but was second in order of the highest organic matter (11%; S1 Table). When considering abiotic variables from multiple regression analysis, we found that $S$ was weakly affected by terrain aspect and marginally by elevation (multiple $R$ = 0.53, aspect $P$ = 0.035, elevation $P$ = 0.05).

Only three species (*O. oppositifolia*, *A. fasciculatum*, and *E. fasciculatum*, i.e., 11% of total $S$) comprised 65% of the total plant cover, and the remaining 26 represented only 35% ($G$ = 0.68; Fig 2C). Plant cover per plot was evenly distributed across the landscape ($G$ = 0.16), and was significantly related to average plant height (0.89–2.42 m, Table 1; $R$ = 0.43, $P$ = 0.009; Table 2), organic matter ($R$ = 0.33, $P$ = 0.04), soil water content ($R$ = 0.37, $P$ = 0.03), and negatively related with aspect ($R$ = -0.37, $P$ = 0.03), but showed no significant relations with AGB and the remaining variables (Table 2 and S3 Table).

## Spatial aggregation of aboveground biomass within the landscape

AGB per plot varied from 0.006 kg m$^{-2}$ to 9.17 kg m$^{-2}$, with a median of 0.69 kg m$^{-2}$ for the study site (ca. seven-ton ha$^{-1}$). Leaf litter on the ground varied from 0.027 kg m$^{-2}$ to 4.76 kg m$^{-2}$, with a median of 0.19 kg m$^{-2}$ (Table 1). Across the landscape, AGB distribution was not homogeneous. The higher AGB plots were located on a few of the lowest elevations, according to the harvesting and the Empirical Bayesian Kriging (Fig 1D). The plot with the greatest biomass (9.17 kg m$^{-2}$, the outlier) was dominated by *Malosma laurina*, but included another four species. The subsequent AGB dominant plots had less than 4 kg m$^{-2}$, including large shrubs such as *O. oppositifolia*, *M. laurina*, and *Q. berberidifolia*. Moreover, 75% of AGB was concentrated in just 30% of the plots (11 plots; $G$ = 0.6; Fig 2B). Without the outlier, inequality in the distribution of AGB also remained high ($G$ = 0.54). Leaf litter had a positive correlation with AGB ($R$ = 0.45, $P$ = 0.006), organic matter ($R$ = 0.43, $P$ = 0.009), soil water content ($R$ = 0.48, $P$ = 0.003), and leaf area ($R$ = 0.56, $P$ < 0.001; S3 Table).

The estimated leaf area per plot ranged from 0.02 to 4.9 m$^2$ m$^{-2}$, with a median of 0.56 m$^2$ m$^{-2}$ (Table 1). 55% of leaf area was concentrated in 22% of the sampling plots (eight plots; $G$ =

0.49, and *G* = 0.42 without the AGB outlier), so leaf area was less concentrated than AGB. In addition, 87% of the total leaf area was distributed in six species (*O. oppositifolia*, *M. laurina*, *E. fasciculatum*, *A. fasciculatum*, *Cneoridium dumosum*, and *Q. berberidifolia*; *G* = 0.71, S4 Table). Leaf area was not correlated with *S* (*R* = 0.19, *P* = 0.2), but was strongly related to AGB (*R* = 0.86, *P* < 0.001), soil water potential (*R* = 0.33, *P* = 0.05), soil organic matter (*R* = 0.72, *P* < 0.001), TDS (*R* = 0.33, *P* = 0.04), soil water content (*R* = 0.62, *P* < 0.001), and dry stem mass (*R* = 0.83, *P* < 0.001; Table 2).

Dry stem mass per plot varied from 0.004 to 8.416 kg m$^{-2}$, with a median of 0.57 kg m$^{-2}$ (S1 Table). It was positively related with soil water potential (*R* = 0.39, *P* = 0.01), but negatively correlated to clay (*R* = -0.35, *P* = 0.03, S3 Table). Dry stem mass represented 86.4% of the total AGB, followed by leaves (12.7%) and reproductive structures (0.8%) from all species (Fig 3A). The same six species that contributed 87% of leaf area also contributed 95% of AGB, mainly due to dry stem mass. These species comprised 28% of the harvested species (*G* = 0.76; Fig 3B). *M. laurina* had the most massive input to AGB (29%), although it had a low relative cover (9$^{th}$ species rank cover, Fig 2C). Dry stem mass per species ranged from 0.003 to 10.5 kg, dry leaf mass from 0.0001 to 1.59 kg, and dry reproductive mass varied from 0.0001 to 0.14 kg. The largest sum of leaf and reproductive mass was from *O. oppositifolia* (S4 Table).

## Patterns of correlation among species richness, aboveground biomass, and landscape metrics

There was a positive relation between relative cover and AGB by species (*R* = 0.59, *P* = 0.005; Fig 3C). Stem dry mass per species had a positive relationship with relative cover (*R* = 0.52, *P* = 0.015), as well as with dry leaf mass per species (*R* = 0.73, *P* < 0.001). Relative cover had a positive relationship with dry reproductive mass per species (*R* = 0.64, *P* = 0.002), dry leaf mass per species (*R* = 0.93, *P* < 0.001), and leaf area per species (*R* = 0.79, *P* < 0.001). Leaf area per species was highly correlated with dry stem mass per species (*R* = 0.81, *P* < 0.001; S5 Table).

AGB was not correlated with *S* (*R* = 0.11, *P* = 0.5; Fig 4A), but it was positively correlated with canopy height (*R* = 0.36, *P* = 0.03), soil water potential (*R* = 0.39, *P* = 0.01; Fig 4B), soil organic matter (*R* = 0.74, *P* < 0.001; S2 Fig), and soil water content (*R* = 0.67, *P* < 0.001; Table 2, S3 Fig), but negatively related with clay (*R* = -0.35, *P* = 0.04; S3 Table). Soil water potential was significantly related to plant height (*R* = 0.46, *P* = 0.004), soil organic matter (*R* = 0.47, *P* = 0.004; Table 2), and pH (*R* = 0.42, *P* = 0.01), as well as with percentage of clay (*R* = 0.50, *P* = 0.04). Soil organic matter was significantly related to vegetation height (*R* = 0.51, *P* = 0.001), soil water content (*R* = 0.82, *P* < 0.001), TDS (*R* = 0.5, *P* = 0.006; Table 2), and surface litter (*R* = 0.43, *P* = 0.009; S3 Table). Although stepwise analysis showed that the full combination of soil water potential, sand, clay, elevation, aspect, slope and topographic index was associated with AGB (*R* = 0.51, *P* = 0.02, AIC = 43.5), the best predictors of AGB were clay and soil water potential (multiple *R* = 0.50, soil water potential *P* = 0.02, clay *P* = 0.04, AIC = 36.1). Metadata per plot and species are shown in S6 and S7 Tables.

## Discussion

By integrating field surveys and geostatistical analysis, we tested the edaphologic and hydrologic correlates of aboveground biomass (AGB) and species richness (*S*) at the local level in a semiarid shrubland. The use of close-to-the-ground remote sensing and geostatistics allowed us to infer *S* and AGB's spatial distributions, and to show how these were affected by landscape heterogeneity or soil properties. Our results indicate that AGB and *S* are strongly clustered in discrete areas of the landscape and highlight the role of water availability as a control on AGB,

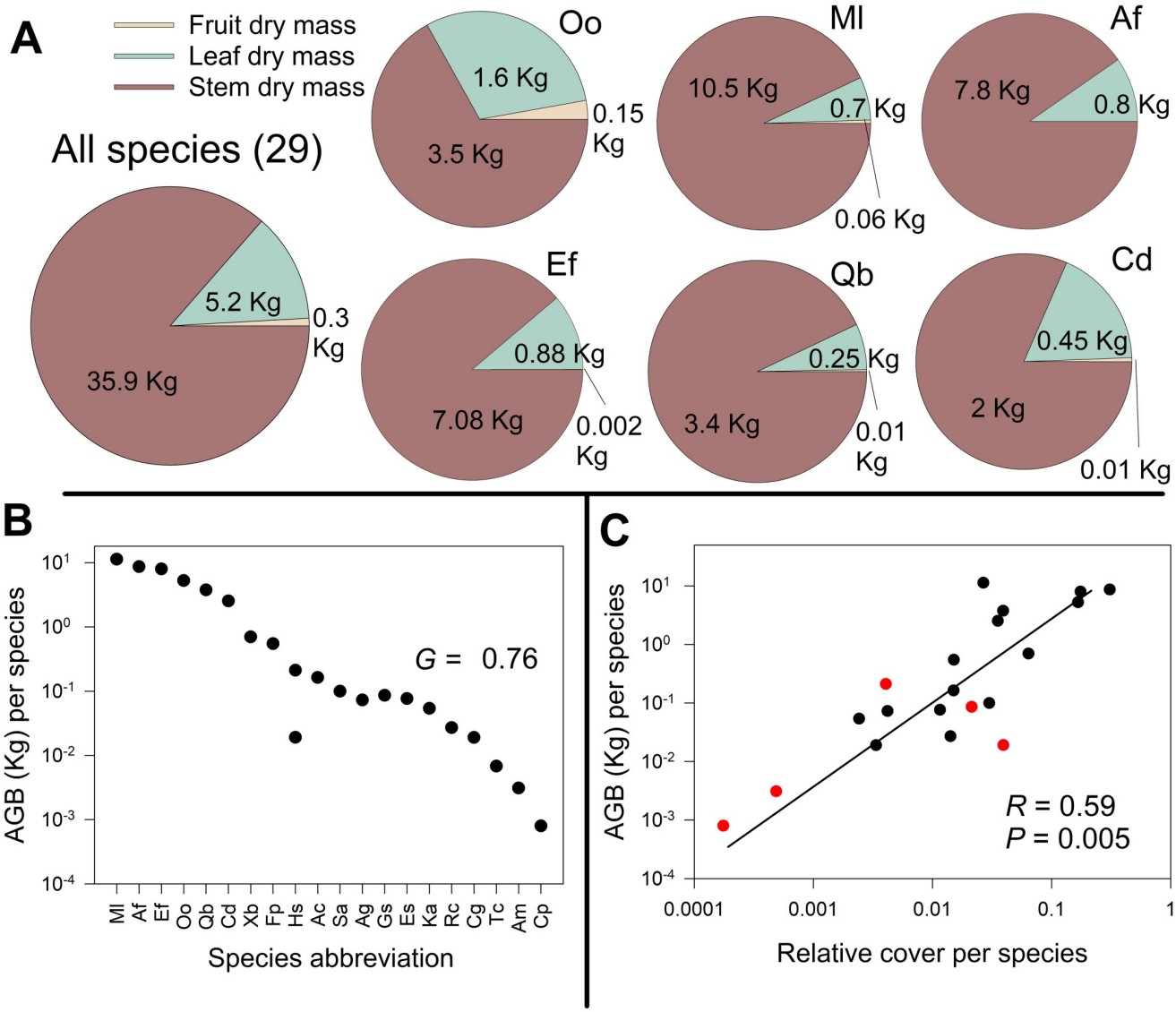

**Fig 3. Aboveground biomass (AGB) distribution between species.** A: Partitioning of AGB (kg) according to organs (stem, leaves, and reproductive structures) for all species and for species with highest AGB. Species abbreviations in S2 Table. B: Distribution of AGB among harvested species. Inequality of AGB per species ($G$ = 0.76). C: Correlation between AGB and the relative cover of each sampled species. Red points are herbaceous species and black points correspond to woody species.

which undoubtedly includes positive feedbacks. This situation was not observed with $S$, which was associated with hillslope aspect and terrain elevation, and lack of extensive shrub cover. The lack of support for any $S$-AGB relation may be due to the influence of different environmental factors on $S$ and AGB, on differences in species size and form, and non-quantified biotic factors controlling $S$.

## Species richness: Patterns of distribution and landscape heterogeneity

Our species pool was comprised of 29 species, some of which (some herbaceous species) may wither early in the dry season. Reduced water availability during the dry season may gradually

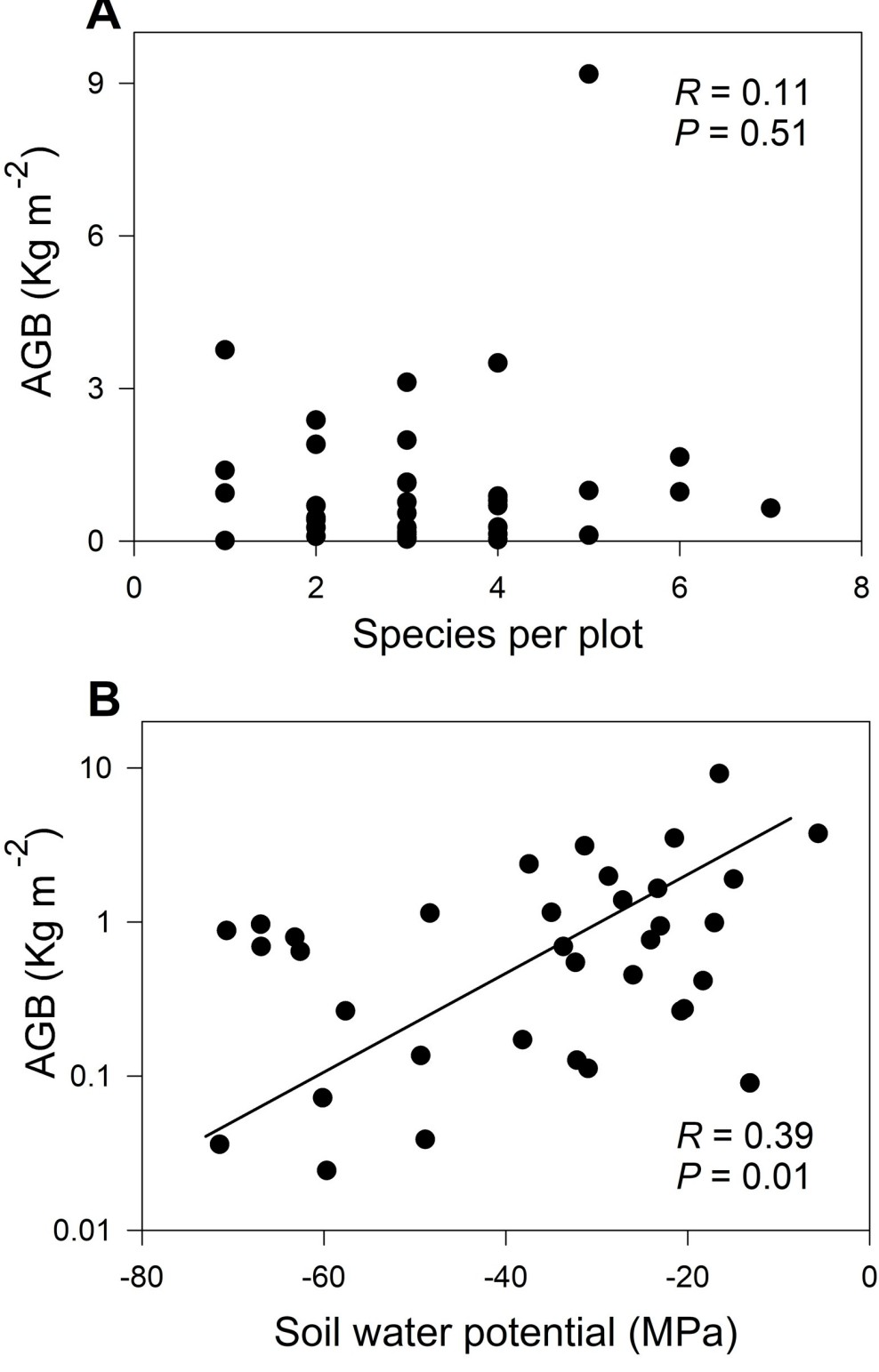

**Fig 4. Species richness and aboveground biomass relationship.** A: Lack of correlation between *S* versus AGB (kg m^-2) for 36 plots. B: Relationship between soil water potential (MPa) and AGB (kg m^-2) per plot at the study site.

diminish *S*, hiding species that cannot withstand environmental stress and competition (e.g. *Marah macrocarpa* and *Chlorogalum parviflorum*) and that only flourish early in the year, as in other sites in California [15]. We showed that terrain aspect was the only abiotic variable associated with the spatial distribution of *S* within this landscape. Four dominant species occupied nearly 71% of our measured vegetation cover, and each of these species entirely covered some plots. However, most species were relatively rare across the landscape. Our results coincide with previous research in chaparral, showing low *S*, also influenced by scale, season and stand age [75, 76].

Cowling et al. [75] affirmed that "few generalizations emerge from the many studies on local diversity in Mediterranean-climate vegetation". These authors attribute the variability and low species diversity found in chaparrals to the diverse types of Californian soils and inter-fire intervals. However, it is a flora that is functionally diverse due to the diversity of adaptations to cope with post-fire regimes, the variety of life-histories and life spans, high permanence of some species as part of the seed bank, and the capacity to resprout following fires, among other characteristics [77]. Similarly, our study site still preserves open ground of 6% to 38% on our small plots, 30 years after the last fire, and *S* was moderately correlated with openness (*R* = 0.38, the inverse of plot cover). This can occur under lower productivity in a drier climate, lower soil N and P [78], perhaps shorter fire intervals [79], and with maintenance of some open space by light pastoral use which has been found to reduce shrub cover in the short term, and at longer terms may reduce the probabilities of fires [80]. Fires and drought interactively shape the composition of chaparrals stands by shrub mortality [81], enhancing the occurrence of herbaceous or pioneering species, which composed 38% of *S* in our study.

Our study finds that landscape geometry affects *S*, even with 50 m spacing of the small plots, reflecting the irregular patchy vegetation patterns commonly found in semiarid areas (e.g., [32, 82]). In other arid sites, water availability in soils is the main driver for canopy cover, because of lower soil evaporation by shade, enhancing the opportunity for seedling establishment close to facilitator trees, promoting density dependent-plant community associations [24]. As shown by many studies in drylands at the scale of our plots, the presence of shrubs is associated with less evaporation due to more shade, less air movement and more litter due to less solar radiation [83], which may have promoted facilitation processes allowing the establishment of rare species. North-facing slopes have higher species richness than south-facing slopes in the watershed that includes Rancho El Mogor [32]. Previous studies have found that higher values of Normalized Difference Vegetation Index (NDVI) are located in north-facing slopes due to a combination of less exposition to solar radiation, which cascades into higher water infiltration in soils, higher vegetation density, and overall greenness (e.g., [32, 34, 84]). As our sampling plots were located on a gentle north-facing slope, we could not discriminate by comparing *S* across hills. Still instead, we found correlations of *S* with terrain aspect despite shallow relief.

The spatial distribution of *S* is related to local topographic heterogeneity in our study site due to terrain aspect and elevation. Other studies have also found that *S* increased with increasing topographic heterogeneity and moderately with soil fertility in California [75]. Topographic heterogeneity may induce different environmental conditions, from different radiation levels to varying runoff intensity, promoting a mosaic of colonization opportunities for local species. However, at different spatial scales, from hundreds to thousands of km$^2$, *S* is mainly associated with water availability [7, 15, 85]. For instance, *S* was associated with relative humidity due to reduced evapotranspiration across an altitudinal gradient of 1700 m in Chile, and in the Spanish Mediterranean Basin, *S* was negatively associated with altitude across a range of 500 m [16].

## Aboveground biomass: Skewed distribution across the landscape

Our results of average AGB per square meter (median value of 0.69 kg m$^{-2}$) were lower than other 30-year-old Californian shrublands with similar annual rainfall (2.7–4.9 kg m$^{-2}$ [4, 86]) and the hillsides of the Sonoran Desert Scrub vegetation (1.146 kg m$^{-2}$; [25]). Even with this relatively low value, the calculation of AGB is probably overestimated, as the most widely used protocols suggest drying biological material for 72 hours at 70 ˚C [65]. However, given that wood has exceptional characteristics of water retention, it is necessary to use a drying temperature of 101–105 ˚C for at least 24 hours [87]. Adopting the new protocol is clearly a priority for studies in all ecosystems with significant abundance of woody plants, to properly represent ecosystem structure and function.

In dry environments at larger scales, precipitation, temperature, and soil texture are the main factors that explain AGB [88]. At the scale of square meters to hectares at our study site, AGB is related to soil water potential, soil organic matter, soil water content, and soil clay content. Our results, therefore, support the hypothesis that AGB of plants from drought-limited regions responds to increased water availability [4, 6, 7] with corresponding supporting structure and higher leaf area index. Moreover, these plants also contribute to soil organic matter and probably to soil moisture in positive feedbacks [27].

AGB correlates with the proportion of clays in soils and soil water potential in our study site. Although our study site is only 17 hectares, its inherent heterogeneity includes varying topography, influencing soil depth. Perhaps in correspondence, we found AGB relationships with soil moisture metrics, such as soil water potential and soil water content, and other indications of soil accumulation, such as clay content. Our higher AGB plots showed higher organic matter content, lower clay content, and higher soil moisture, which is coincident with other studies [89]. Vegetation presence can create further feedbacks to this causal relationship, as larger plants in this environment may develop deeper root systems, which could promote hydraulic redistribution to topsoil layers [26, 90] and probably deeper development of the soil. Unfortunately, soil depth has not been mapped at our site and surveys have been frustrated by shallow stoniness. Nearby outcrops suggest that fissuring of the rock underlying our site, which could provide scattered water reserves [91].

In our study site, *Malosma laurina* had the highest proportion of AGB per species, although its relative cover was not the largest (S1 Table). It was also one of the tallest and widest spreading shrub species along with *Ornithostaphylos oppostifolia* (the species with the highest relative cover, S2 Table). *M. laurina* had the largest proportion of AGB per species due to its more massive stems. Given that *M. laurina* and *Quercus* spp. are able to resprout after fires and being likely to suffer less droughts than species like *Adenostoma fasciculatum* or *Ceanothus* spp. [92] due to their deeper root systems, they may achieve larger relative growth rates and higher survival probability than other species.

## Lack of support for *S* and AGB relationships at the local scale in Mediterranean shrublands

In our study, water availability determines AGB, but not *S*. We found a positive relation of AGB with soil water potential as identified in other drylands [27], and non-significant associations between *S* and AGB within the study site [93]. Diversity ranged from one to seven species at the plot-scale (ca. 12.5 m$^2$), and species were highly unequal in relative abundance, as three species were disproportionately dominant. Given the variability in individual size among species, AGB was also unequal and unrelated with *S*. For example, in the plots harboring the highest species richness, we found AGB of 0.64 kg m$^{-2}$; for those plots having six species, there was 0.96 and 1.65 kg m$^{-2}$, and for those of five species, we found 0.11, 0.98, and 9.17 kg m$^{-2}$, the

highest AGB plot. In other sclerophyllous forests, *S* and functional richness promote higher wood production [94], thus implying complementarity among species to increase AGB. However, at the scale of small plots, our data did not show such a relationship. We showed that different abiotic controls determine the lack of associations between *S* and AGB within this chaparral, at the range of square meters to hectares. In fact, disturbance of AGB accumulation may favor *S*. A plausible factor favoring *S*, beyond the scope of our study, may be that disturbance from feeding and activity by small (rabbits, squirrels) or large (deer, cattle) mammals limits dominance and promotes heterogeneous microsite conditions, favoring richness [95].

We also found that the distribution of AGB and leaf area were highly unequal among plots and species (*G* = 0.6 and 0.76 for AGB, respectively, and *G* = 0.49 and 0.71 for leaf area by plots and species, respectively), but highly correlated with each other. Although plant cover was much more homogeneous (*G* = 0.16), our results suggest AGB accumulation, and to a lesser extent leaf area, were very heterogeneous across the site, perhaps analogous to islands of fertility in other ecosystems (e.g., [96]). Two shrub species (*Ornithostaphylos oppositifolia* and *Eriogonum fasciculatum*) covered 30 and 18% of total plant cover in our study site. Four other species occupied nearly 71% of the total vegetation cover, and each of these species entirely covered some plots, generating a complex mosaic of species richness across the landscape (Fig 1). In consequence, in our study site, AGB patterns seems to be influenced by a few dominant species, a result according to the mass-ratio hypothesis [8].

## Conclusions

Our study shows that different biotic and abiotic conditions correlated with AGB and *S* within a 17 ha site of semi-arid shrubland in Baja California, México. Both aboveground biomass (mostly stem mass) and species richness were highly clumped across the landscape but in different areas. The substantial spatial heterogeneity of these ecological properties can be partially attributed to various environmental controls: AGB is most strongly linked to water availability, and *S* to terrain heterogeneity and vegetation dynamics, in particular, to plant cover, hillslope aspect, and elevation. Whether these associations are pervasive in shrublands across the larger watershed and the bioregion, at similar scales and different locations, remains to be seen.

## Supporting information

**S1 Appendix. Script to obtain the regression stepwise analysis.**
(TXT)

**S2 Appendix. Script to obtain the level of inequality via ´ineq´ package.**
(TXT)

**S1 Table. Data obtained by plot.**
(CSV)

**S2 Table. Table of species abbreviation (Abb) list and their contribution with relative cover, life form, life cycle, height in meters (H), relative stem mass, relative leaf area, frequency and plant habit.** NA = not available, the date that could not be encountered, or species that were seen outside the plot.
(DOCX)

**S3 Table. Correlation matrix of all measurements taken by plot.**
(XLSX)

**S4 Table. Data obtained by species.**
(CSV)

**S5 Table. Correlation matrix of all measurements taken by species.**
(CSV)

**S6 Table. Metadata from the data obtained by plot.**
(XLSX)

**S7 Table. Metadata from the data obtained by species.**
(XLSX)

**S1 Fig. Soil texture triangle of 36 plots across 17 hectares of semiarid shrubland in Rancho El Mogor, Baja California, México.**
(TIF)

**S2 Fig. Map of interpolation of species richness (*S*), aboveground biomass (AGB), soil water potential (water potential) and organic matter (OM) of 36 sampling plots at Rancho El Mogor, Baja California, México.**
(TIF)

**S3 Fig. Map of interpolation of species richness (*S*), aboveground biomass (AGB), shrub cover (plant cover) and soil water content of 36 sampling plots at Rancho El Mogor, Baja California, México.**
(TIF)

## Acknowledgments

This study was performed by SDDL as a partial pre-requisite for a doctoral degree in Life Sciences at the Posgrado de Ciencias de la Vida, CICESE. We thank N. Badan-Dangón for providing access and facilities at the Rancho El Mogor. We acknowledge field support from E. López, M. Salazar, L. Tellechea, R. Santos-Cobos, Y. Romero-Toledo, J. L. Sánchez-Dahlinger, P. López-Sarmiento, and E. Pérez-Robles. Comments on previous versions and advice on analysis was provided by F.W. Ewers, and S. Ceccarelli. We appreciate the comments by E. Alvarez Davila and two anonymous reviewers on a previous version. ERV thanks the U.S. Fulbright-Garcia Robles program for providing support during his sabbatical period at CICESE.

## Author Contributions

**Conceptualization:** Rodrigo Méndez-Alonzo.

**Data curation:** Samantha Díaz de León-Guerrero.

**Funding acquisition:** Rodrigo Méndez-Alonzo.

**Investigation:** Samantha Díaz de León-Guerrero, Rodrigo Méndez-Alonzo.

**Methodology:** Rodrigo Méndez-Alonzo, Stephen H. Bullock, Enrique R. Vivoni.

**Software:** Samantha Díaz de León-Guerrero.

**Supervision:** Rodrigo Méndez-Alonzo, Stephen H. Bullock.

**Validation:** Stephen H. Bullock, Enrique R. Vivoni.

**Writing – original draft:** Samantha Díaz de León-Guerrero, Rodrigo Méndez-Alonzo.

**Writing – review & editing:** Stephen H. Bullock, Enrique R. Vivoni.

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
