## [Decision Letter · Decision Letter 0]

23 Feb 2021

PONE-D-20-36492

Hydrological correlates of biomass and species richness in a Mediterranean-climate shrubland

PLOS ONE

Dear Dr. Rodrigo Mendez-Alonzo,

Thank you for submitting your manuscript to PLOS ONE. After careful consideration, we feel that it has merit but does not fully meet PLOS ONE’s publication criteria as it currently stands. Therefore, we invite you to submit a revised version of the manuscript that addresses the points raised during the review process.

We look forward to receiving your revised manuscript.

Kind regards,

Wang Li

Academic Editor

PLOS ONE

Journal Requirements:

"Funding: this work was supported by CONACYT Doctoral scholarship (274874), CONACYT487

INEGI (278755), and CICESE (681-117)."

"RMA 278755 Fondo Sectorial CONACYT INEGI

Website: https://www.inegi.org.mx/investigacion/conacyt/default.html

NO: The funders had no role in study design, data collection and analysis, decision to publish, or preparation of the manuscript."

3. We note that Figure 1 in your submission contain map images which may be copyrighted. All PLOS content is published under the Creative Commons Attribution License (CC BY 4.0), which means that the manuscript, images, and Supporting Information files will be freely available online, and any third party is permitted to access, download, copy, distribute, and use these materials in any way, even commercially, with proper attribution. For these reasons, we cannot publish previously copyrighted maps or satellite images created using proprietary data, such as Google software (Google Maps, Street View, and Earth). For more information, see our copyright guidelines: http://journals.plos.org/plosone/s/licenses-and-copyright.

3.1.    You may seek permission from the original copyright holder of Figure 1 to publish the content specifically under the CC BY 4.0 license. 

3.2.    If you are unable to obtain permission from the original copyright holder to publish these figures under the CC BY 4.0 license or if the copyright holder’s requirements are incompatible with the CC BY 4.0 license, please either i) remove the figure or ii) supply a replacement figure that complies with the CC BY 4.0 license. Please check copyright information on all replacement figures and update the figure caption with source information. If applicable, please specify in the figure caption text when a figure is similar but not identical to the original image and is therefore for illustrative purposes only.

Reviewers' comments:

Reviewer's Responses to Questions

**Comments to the Author**

1. Is the manuscript technically sound, and do the data support the conclusions?

Reviewer #1: Yes

Reviewer #2: Partly

Reviewer #3: Yes

2. Has the statistical analysis been performed appropriately and rigorously? 

Reviewer #1: Yes

Reviewer #2: Yes

Reviewer #3: Yes

3. Have the authors made all data underlying the findings in their manuscript fully available?

Reviewer #1: Yes

Reviewer #2: Yes

Reviewer #3: Yes

4. Is the manuscript presented in an intelligible fashion and written in standard English?

Reviewer #1: Yes

Reviewer #2: Yes

Reviewer #3: Yes

5. Review Comments to the Author

Reviewer #1: The authors studied a chaparral ecosystem to understand drivers of aboveground biomass and plant species richness, and their relationship. The authors found a positive relationship between aboveground biomass and water availability, while species richness covaried with landscape properties and reduced shrub cover. These are interesting findings, although the positive effect of increased water availability on biomass is no surprise, as well as the evidence for spatial heterogeneity of physical environment increasing the importance of species richness. I further suggest in my comments, some analyses and illustrations that would enhance the scope of the paper.

Reviewer #2: This is a valuable study of an important region and it would fit in this journal. The authors put a good deal of time in the field investigation, I have several comments on the result and discussion part. This paper should be accepted with major revisions.

1) In the result part, the description on landscape and environmental characteristics (Lines 230-237) is suggested to put into the method part.

2) The rest of the results section should be divided into three or four sections, and each with a subtitle.

3) Accordingly, the discussion part should also be done according to three or four sub themes in the result part.

Reviewer #3: In general, León-Guerrero et al. carried out an original analysis and present novel results on the effect of local scale environmental gradients on vegetation in semi-arid regions. However, I have some comments regarding the methods, which may affect the results. I recommend adjusting the introduction and discussion to make these concerns visible.

a) The emphasis from the beginning on the relationship between plant productivity and aerial biomass is not relevant to the study. 

In lines 91-93, the authors state that aboveground biomass is a PROXY of plant productivity and rely on the publication of Alder et al. (2011). However, the Alder et al. (2011) study uses peak annual biomass, which can be an efficient measure of productivity in herbaceous communities, especially when herbivory is low. Given that in the community studied by León-Guerrero et al. the woody component has 80% of the biomass and has been affected by grazing, the quote does not really justify anything. 

On the other hand, studies indicate that the relationship between biomass-productivity can be highly variable in the same site, from negative to positive or none, depending on the age of the plants, successional stage, herbivory, senescence or type of disturbance (Terhorst and Munguia 2008). Other studies in tropical forests show that the relationship between productivity and biomass is not clear (Keeling and Phillips, 2007). 

b) The methods used leave doubts about the accuracy of biomass estimates. First, the plots used to quantify species richness and vegetation cover were 12.5 m2, but biomass was measured in one square meter located at the northern edge of each plot. This may generate biases since biomass is spatially distributed in patches. It would have been important to have several 1 m2 samples for destructive harvesting within each plot, rather than just one. Another alternative is the use of allometric equations for shrubs in semi-arid zones that are based on crown measurements of individuals (Conti et al. 20013). Other equations for shrubs and herbs can be found in Alamgir & Al-Amin (2008). In other words, it is possible to find in the literature models that allow estimating plot biomass with less uncertainty.

Finally, the biomass samples were dried at 70 oC, which can lead to overestimation of biomass, especially woody biomass. According to Williamson and Wiemann (2010), kiln drying requires temperatures above 100 oC, because wood contains bound water and free water that cannot be fully expelled at lower temperatures. 

• Adler PB, Seabloom EW, Borer E 538 T, Hillebrand H, Hautier Y, Hector A, et al. Productivity is a poor predictor of plant species richness. Science (80- ). 2011;333(6050).

• Alamgir, M., & Al-Amin, M. (2008). Allometric models to estimate biomass organic carbon stock in forest vegetation. Journal of forestry research, 19(2), 101.

• Conti, G., Enrico, L., Casanoves, F. et al. Shrub biomass estimation in the semiarid Chaco forest: a contribution to the quantification of an underrated carbon stock. Annals of Forest Science 70, 515–524 (2013). https://doi.org/10.1007/s13595-013-0285-9)

• Keeling, H. C., & Phillips, O. L. (2007). The global relationship between forest productivity and biomass. Global Ecology and Biogeography, 16(5), 618-631.

• TERHORST, C.; MUNGUIA, P. Measuring ecosystem function: consequences arising from variation in biomass-productivity relationships. Community Ecology, 2008, vol. 9, no 1, p. 39-44.

• Williamson, G. B., & Wiemann, M. C. (2010). Measuring wood specific gravity… correctly. American Journal of Botany, 97(3), 519-524.

6. PLOS authors have the option to publish the peer review history of their article (what does this mean?). If published, this will include your full peer review and any attached files.

Reviewer #1: No

Reviewer #2: No

Reviewer #3: Yes, Esteban Alvarez-Davila

---

## [Author Response · Author response to Decision Letter 0]

30 Apr 2021

Response to Editor and Reviewers MS PONE-D-20-36492

Diaz de Leon et al. "Hydrological correlates of biomass and species richness in a Mediterranean-climate shrubland"

Dear Prof. Wang Li, 

Academic Editor

PLOS ONE

Many thanks for allowing us the opportunity to re-submit a corrected version of our manuscript Attached to this e-mail, please find the following items:

A letter that responds to each point raised by the academic editor and reviewer(s). 

Reviewers Response PONE-D-20-36492.docx 

A marked-up copy of our manuscript.

DiazdeLeon PONE-D-20-36492 R1 TRACK CHANGES.docx

An unmarked version of our revised paper.

DiazdeLeon PONE-D-20-36492 R1 CLEAN.docx

We expect this new version will respond to the concerns raised by the Editor and reviewers, and fulfill the criteria of quality expected for PLoS ONE.

Sincerely,

Rodrigo Méndez-Alonzo

Corresponding author

 

Journal Requirements:

Response: We have double-checked that our MS complies with the templates.

"Funding: this work was supported by CONACYT Doctoral scholarship (274874), CONACYT487

INEGI (278755), and CICESE (681-117)."

We note that you have provided funding information that is not currently declared in your Funding Statement. However, funding information should not appear in the Acknowledgments section or other areas of your manuscript. We will only publish funding information present in the Funding Statement section of the online submission form. Please remove any funding-related text from the manuscript and let us know how you would like to update your Funding Statement. 

Response: We have deleted the statements regarding funding information from the Acknowledgments section.

Response: Many thanks for modifying our Funding statements within the online submission system. The amendments are (now included in the cover letter):

"RMA 278755 Fondo Sectorial CONACYT INEGI

Website: https://www.inegi.org.mx/investigacion/conacyt/default.html

SDDLG 274874 CONACYT Scholarship for PhD students

NO: The funders had no role in study design, data collection and analysis, decision to publish, or preparation of the manuscript."

3. We note that Figure 1 in your submission contain map images which may be copyrighted. All PLOS content is published under the Creative Commons Attribution License (CC BY 4.0), which means that the manuscript, images, and Supporting Information files will be freely available online, and any third party is permitted to access, download, copy, distribute, and use these materials in any way, even commercially, with proper attribution. For these reasons, we cannot publish previously copyrighted maps or satellite images created using proprietary data, such as Google software (Google Maps, Street View, and Earth). For more information, see our copyright guidelines: http://journals.plos.org/plosone/s/licenses-and-copyright.

Response: Many thanks for noting this. We have changed our Figure 1 to include as background an image from USGS National Map Viewer, not subject to copyright (also included in Supplementary figures S2 and S3 Fig)

Review Comments to the Author

Reviewer #1: The authors studied a chaparral ecosystem to understand drivers of aboveground biomass and plant species richness, and their relationship. The authors found a positive relationship between aboveground biomass and water availability, while species richness covaried with landscape properties and reduced shrub cover. These are interesting findings, although the positive effect of increased water availability on biomass is no surprise, as well as the evidence for spatial heterogeneity of physical environment increasing the importance of species richness. I further suggest in my comments, some analyses and illustrations that would enhance the scope of the paper.

R: We appreciate the reviewer’s opinion, and expect this new version will respond to her (his) concerns.

Reviewer #2: This is a valuable study of an important region and it would fit in this journal. The authors put a good deal of time in the field investigation, I have several comments on the result and discussion part. This paper should be accepted with major revisions.

R: We appreciate the reviewer’s opinion, and expect this new version will respond to her (his) concerns.

1) In the result part, the description on landscape and environmental characteristics (Lines 230-237) is suggested to put into the method part.

R: We appreciate the suggestion by the reviewer. The landscape characteristics (elevation, aspect, and slopes) are now in Lines 162-164. We maintain the results from our quantification of soil properties in the results section, as these were novel measurements from our study and allow the readers to understand further results.

2) The rest of the results section should be divided into three or four sections, and each with a subtitle.

R: Many thanks for this suggestion. Results section now divided in four sections.

3) Accordingly, the discussion part should also be done according to three or four sub themes in the result part.

R: Many thanks for this suggestion. Discussion section now divided in three sections.

Reviewer #3: In general, León-Guerrero et al. carried out an original analysis and present novel results on the effect of local scale environmental gradients on vegetation in semi-arid regions. However, I have some comments regarding the methods, which may affect the results. I recommend adjusting the introduction and discussion to make these concerns visible.

a) The emphasis from the beginning on the relationship between plant productivity and aerial biomass is not relevant to the study. 

In lines 91-93, the authors state that aboveground biomass is a PROXY of plant productivity and rely on the publication of Alder et al. (2011). However, the Alder et al. (2011) study uses peak annual biomass, which can be an efficient measure of productivity in herbaceous communities, especially when herbivory is low. Given that in the community studied by León-Guerrero et al. the woody component has 80% of the biomass and has been affected by grazing, the quote does not really justify anything. 

R: We appreciate the reviewer’s observation. In response, we have re-written our Introduction, to be more focused on arid environments, and deleted the reference to the Adler et al study from this section.

On the other hand, studies indicate that the relationship between biomass-productivity can be highly variable in the same site, from negative to positive or none, depending on the age of the plants, successional stage, herbivory, senescence or type of disturbance (Terhorst and Munguia 2008). Other studies in tropical forests show that the relationship between productivity and biomass is not clear (Keeling and Phillips, 2007). 

R: We agree with the reviewer, as several ecosystem processes may interfere with the observation of any biomass-species richness association. We now review the matter specifically for drylands, in lines 63-72.

b) The methods used leave doubts about the accuracy of biomass estimates. First, the plots used to quantify species richness and vegetation cover were 12.5 m2, but biomass was measured in one square meter located at the northern edge of each plot. This may generate biases since biomass is spatially distributed in patches. It would have been important to have several 1 m2 samples for destructive harvesting within each plot, rather than just one. 

R: We apologize for not being clear enough on this point. 1. We agree with the reviewer that biomass is aggregated across the landscape, and we quantified this aggregation by using the Gini coefficient (G, an econometric indicator of inequality and accumulation of data regularly used for wealth comparisons across nations). Using this metric allowed us to directly quantify the intensity of aggregation of AGB within this landscape. This index has been used previously to quantify skewness in plant size (Dixon et al. 1987). We have clarified this point in the Methods section (L. 236-242). Our calculations of G are presented in the Results section (L. 306-308 and Fig. 2B), showing the spatial aggregation and heterogeneity in biomass.

2. Although we had one AGB plot per each plot, overall we had 36 biomass plots. This sample size allowed us to interpolate the patterns of biomass accumulation across the landscape, as shown in our figures 1, S2, and S3. In addition, the sample allowed us to discuss bias of biomass by species.

Dixon, P. M., Weiner, J., Mitchell-Olds, T., & Woodley, R. (1987). Bootstrapping the Gini coefficient of inequality. Ecology 68(5), 1548-1551.

Another alternative is the use of allometric equations for shrubs in semi-arid zones that are based on crown measurements of individuals (Conti et al. 20013). Other equations for shrubs and herbs can be found in Alamgir & Al-Amin (2008). In other words, it is possible to find in the literature models that allow estimating plot biomass with less uncertainty.

R: We appreciate the concern raised by the reviewer. We opted not to use allometric equations for this study because multi-species equations would require validation for each species (as shown in Conti et al. 2013), before applying in our study. The validation would require an experimental design and data processing apart from our study.

Finally, the biomass samples were dried at 70 oC, which can lead to overestimation of biomass, especially woody biomass. According to Williamson and Wiemann (2010), kiln drying requires temperatures above 100 oC, because wood contains bound water and free water that cannot be fully expelled at lower temperatures. 

R: We agree with this observation by the reviewer, that the calculation of the specific gravity of wood requires drying samples at 101-105 C to evaporate bound water from wood. In our samples to determine aboveground biomass, we had a combination of leaves, fruits and wood, and thus we followed the protocols of Perez-Harguindenguy et al. (2013), who recommended oven-drying samples at 70 C for 72 hours (page 207, Perez-Harguindenguy et al. 2013, and corrigendum in Australian Journal of Botany, 2016, 64:715–716), considering the calculation of specific wood gravity only a special case to employ temperatures of 101-110 C. 

As the reviewer correctly points, this procedure would overestimate wood biomass, which was clearly the great majority our samples. We now indicate this over-estimation in the Discussion section, suggesting that new protocols worldwide convey the criterion of 101-105 C (L. 424-430).

• Adler PB, Seabloom EW, Borer E 538 T, Hillebrand H, Hautier Y, Hector A, et al. Productivity is a poor predictor of plant species richness. Science (80- ). 2011;333(6050).

• Alamgir, M., & Al-Amin, M. (2008). Allometric models to estimate biomass organic carbon stock in forest vegetation. Journal of forestry research, 19(2), 101.

• Conti, G., Enrico, L., Casanoves, F. et al. Shrub biomass estimation in the semiarid Chaco forest: a contribution to the quantification of an underrated carbon stock. Annals of Forest Science 70, 515–524 (2013). https://doi.org/10.1007/s13595-013-0285-9)

• Keeling, H. C., & Phillips, O. L. (2007). The global relationship between forest productivity and biomass. Global Ecology and Biogeography, 16(5), 618-631.

• TERHORST, C.; MUNGUIA, P. Measuring ecosystem function: consequences arising from variation in biomass-productivity relationships. Community Ecology, 2008, vol. 9, no 1, p. 39-44.

• Williamson, G. B., & Wiemann, M. C. (2010). Measuring wood specific gravity… correctly. American Journal of Botany, 97(3), 519-524.

REVIEWER B Comments:

The authors studied a chaparral ecosystem to understand drivers of aboveground biomass and plant species richness, and their relationship. The authors found a positive relationship between aboveground biomass and water availability, while species richness covaried with landscape properties and reduced shrub cover. These are interesting findings, although the positive effect of increased water availability on biomass is no surprise, as well as the evidence for spatial heterogeneity of physical environment increasing the importance of species richness. I further suggest in my comments, some analyses and illustrations that would enhance the scope of the paper.

Title

Lines5-6: Should title instead be something like “Determinants of biomass and species richness in a Mediterranean-climate shrubland” since species richness best correlated with other variables?

R= We thank the reviewer for this suggestion. Following her (his) advice, we changed the title to “Hydrological and topographic determinants of biomass and species richness in a Mediterranean-climate shrubland”

Introduction

Line 62-63: state clearly that water availability is one of the main factors influencing plant species richness. Other factors, such as community assembly factors (e.g., seed propagule) or disturbance, could also influence plant species richness.

R= This sentence is now in Line 60-62, and we added other factors that have been found to contribute plant species richness (Huston, 1979; Bassa et al., 2012; Xu et al., 2016).

Lines 64-65: The connection between aboveground net productivity and precipitation to plant species richness is not clear in this sentence. Perhaps authors meant to say something like “Although, plant species richness has been related to several ecosystem functions in drylands, association to productivity remains unclear”.

R= We thank the reviewer for this clarification, now in Line 63-67.

Lines 75-76: What does “Positive-reinforcementprocesses” mean in this context? 

R=We apologize for not being clear enough on this point. We explain further on Line 74-78. This positive reinforcement process has been explained by Scanlon et al. (2007) as positive spatial feedbacks, where the probability of establishment increases with more tree density, like water availability which is the main driver of plant establishment, but also the canopy itself helps to maintain soil moisture because of more shade, leading to reduced bare soil evaporation.

Line 93-95: including facilitative relationships in some cases. See 

Species diversity enhances ecosystem functioning through interspecific facilitation

BJ Cardinale, MA Palmer, SL Collins

Nature 415 (6870), 426-429

R=We added a comment about the facilitation process from species diversity in the previous paragraph (Line 83-85) that refers to the positive reinforcement process like shading and less soil evaporation at more canopy cover, permitting less drought-tolerant species to coexist, therefore, having more diversity.

Lines 103-104: replace “tests” for “studies” and add “in this system” after abiotic factors. Also, authors mention that the relationship between AGB and S is not fully understood, but what is known in this system?

R=We restructured the sentence to explain further the need for studying this type of vegetation. (line 106-108). We explained what is known in systems similar to our study site in lines 88-91.

Line 118: add reference that supports hypothesis

R=Thank you for the observation, we have added two references supporting the hypothesis that the relationship of species richness and biomass may be influenced by water availability (Line 59-60 and 108-111 Including Li et al 2013).

Lines 118-122: and what is the hypothesis for these other variables with the relationship between S-ABG, and for each (S and ABG)?

R=To clarify this section, we have re-written the final paragraphs of our introduction to establish a set of specific hypotheses. Please find the new hypotheses on lines 55-60.

Line 222: I didn’t see variable scaling in the authors code (see S2 Appendix stepwise analysis Data plot.txt), considering that variables had different units. Does variables scaling influence the outcome of the results?

R= Following reviewer’s suggestion, we ran the analyses again with normalized data, to check if there was any effect of data scaling. P values and multiple R2 remained the same with normalized data. Appendix S1 now includes normalized data.

Lines 122-123: how will generated maps of AGB and S by interpolation techniques support statistical analysis in this study? 

R= We consider the interpolation maps to be visual aids that allow better comprehension of the statistical analysis. We also recognize that these sentences were confusing, and they have been deleted from the introduction. Please check our re-written introduction, with a new set of predictive hypotheses and more theory to predict the role of species in ecosystem functioning.

Methods

Line 127: In figure 1, authors refer to 36 sampling plots and later in line 158 first mention them in the methods, however, it is unclear why 36 plots were chosen.

R= We clarified this important point in Lines 159-161. We selected the 36 plots because our study area was a polygon of ca. 17 hectares. To cover this area homogeneously, we first generated a grid of 50 x 50 m using GIS. The number of intersections on the grid was 36. The selected area (17 hectares) is a natural preserve within a private property, and corresponds to a geomorphic unit and to the footprint area for an eddy covariance tower.

Lines 132-134: Because this system has such marked seasons, can authors provide means for precipitation broken down by dry and wet seasons within the same time range (1986-2018)? Time range should include the period of observations.

R= Published data and open databases are not up-to-date. We have changed the citation to one with a slightly older record (1980-2009) that specifies the mean annual precipitation and also monthly precipitations in the rainy winter season (18-63 mm), as well as the dry summers (1-6 mm) (León et al., 2014) now on Line 129-131.

Line 138: in “has been sporadically used by cattle”, did cattle have access to the plots for grazing? If so, it should be acknowledged that study plots were eventually grazed since this can influence the outcome of plant richness and biomass.

R= We have improved the statement as possible, and added the acknowledgement: the site “has been traversed or browsed a few days of the year by a small herd of cattle inclined to forage in surrounding areas that are more verdant or tended.” It is possible they may have had patchy affects on openness and species composition, particularly in the early post-fire years, now inlines 134-136. 

Lines 138-139: clarify why this information is relevant.

R= We added that this tower, which also helped delimit our study zone because there are previous landscape studies (Line 136-138).

Line 146: why was initial mapping done and how often? Was it for site characterization in table 1? If so, say this up front in this paragraph.

R= It was done only once, at the very beginning of the study to delimit our study area and mark the plots, now in Line 147-151.

Lines 161: Explain why only one observation was performed. Is there any previous knowledge of the system (e.g., no change in species composition within and across seasons) that would justify one observation in April?

R= In our study site, November to April is the period of the year, during the rainy season, when most growth occurs, as documented from satellite imagery (Del Toro-Guerrero et al. 2019) and known by more than twenty years of experience of some of our personnel. Protocols indicate that “biomass must be sampled at peak of the growing season and no less than 3 mo disturbance-free … i.e. no mowing, haying, grazing or fire. It is important to point out that prior disturbance, even if it is sustained and/or intense, is irrelevant in terms of site selection”. (Fraser et al., 2014). Now described in L. 166-167, 176-177, 180-182.

Lines 162-164: I am not familiar with this technique to assess plant cover. How precise were authors in determining plant cover by using images obtained 5m from the ground? Especially for short plants and considering various layers of the vegetation. If available, provide citations for the use of this technique and explain how species-specific cover was quantified from images.

R= The set of photographs taken at 5 m height allowed us to calculate the shrub canopy area more precisely than the conventional calculations of from two or three dimensions (e.g. major and minor axes, Conti et al. 2013). Using Image J as image processing software, we measured canopy areas directly on the photographs. Also, the resolution and colors of the images allowed us to discriminate among species. As the reviewer correctly points, undercanopy plants could not be quantified (but are uncommon in chaparral). The idea to use this method was obtained from intrasite photography in archaeology, were mast-photography is a widely-used method (reference now provided in paragraph). Now in Lines 170-176. 

Line 168: nomenclature for what? Clarify this in the text.

R= Now in Line 176-177, we clarify that nomenclature for plant taxa was based on the recent authoritative publication (cited).

Line 170: clarify why AGB was harvested in February 2018 and in how many replicates. Was February the peak of biomass for this system?

R= Now in Line 180-182, we state that biomass sampling must be done near the peak biomass of the year (Fraser et al., 2014), before the end (March or April) of the rainy season at our site. We also want to specify that we did not make replicate subsamples of biomass per plot, relying on destructive sampling of n=36 plots of 1 m2. We did not establish any hypotheses concerning intra-plot variability, as this would involve different research questions and experimental design and methods.

Line 173-174: clarify what authors meant by “species origin ofthe remains”. Was it dead biomass?

R= Yes, the litter was dead leaves, twigs and wood fragments, almost always identifiable to species (Line 185-186). 

Lines 178-182: Do the authors have any information about how abundant (average cover) these groups species are and if their absence in the harvested plots would influence sampled AGB values?

R= The information on the relative cover of every species we measured is now given by species in Tables S2 and S4. The herbaceous, parasitic and vines species would be trivial in individual and total biomass. The shrubs are persistent, but those species are small (or spindly and semi herbaceous) and were very patchy and of low contribution to cover on our site. Their absence in the harvested plots was not surprising or a cause for concern about AGB results. 

Lines 206-208: how did authors get to the best estimator of plant richness? 

R= Now in Line 218-224. We based our choice of estimators on the recommendations in Colwell and Coddington (1994).

Results

It would be veryhelpful and it would add to the paper if authors could add other layers to the map in Figs 1C and 1D for the distribution of key soil properties and shrub cover – i.e. key variables driving AGB and S. For example, mark areas in the map where soil watercontent was highand low.

R= We thank the reviewer for this suggestion. In response, we now include new maps of soil organic matter and soil water potential (Figure S2), as well as plant cover and soil water content (Figure S3).

Line 310: there’s only mention to dry fruit mass, were all or most of the species in this reproductive stage? If so, how much information in terms of biomass and greenness was potentially missed if vegetation sampling was done past flowering? I would expect peak of greenness or productivity to occur prior reproductive events. This would be related to resource accumulation for investment in reproductive structure. Or does this does not apply to this system? Please clarify.

R= Now we use the term “reproductive dry mass”, to include flowers, flower abortions, and fruits, ranging from 0.0001 to 0.14 kg per sp among the 36 sampling sites (in Table S4), and 0.056 kg per m2 combining all species per plot (in Table S2). We do not have precise information on the reproductive phenology of each species, but there are notable differences. One or two common species were probably underrepresented in reproductive dry mass. And the interannual variation of reproduction in some species is notorious (and as yet unquantified). Of course, annual carbon allocation budgets would be far beyond our scope. However, it is apparent that reproductive mass is minor compared to leaves and stems. 

Discussion

Lines 453:355: not sure if authors are trying to say this, but reduced species number could also be part of species phenology, meaning that the timing of activity occurs in the wet season and plants senesce before the dry season. This makes me wonder if ‘filtered species’ could have contributed significantly to the species number during measurements. 

R= We agree that phenology is important, as commented also in Methods. We have modified the text (Lines 375-378) and noted two examples. 

Lines 371-372: is there any influence of cattle grazing here? Grazing normally increases species richness. If so, how grazed is this study site?

R=We comment on Lines 472-475 on the significance of “open” ground (with various references), some effected by cattle (probably abetted by rabbits, squirrels, and physical forces!). We commented on the cattle incursion on Lines 134-136. There are no data, historic or recent, with precise quantification on incursion (to an area of c. 50 ha), so our comments in the text are about as much as can be said.

Lines 434 and line 443: a suggestion to better explore AGB-S relationship: because unequal relative abundance is so marked in this study site, authors would benefit from using Community Weighted Means (CWMs) at the plot-level. This analysis wouldweight the relative abundance of species into AGB and then authors could test the CWM AGB-S relationship. See package FD, function functcompin R.

R= We appreciate the suggestion by the reviewer. In our design, we established area-based measurements. However, we do not have biomass for replicated single shrubs per species, and thus we are unable to calculate CWM. We did quantify the skewness of the distribution of biomass and species richness using the Gini index of inequality (in this case of plots, for S and AGB). This metric allowed us to show the intensity of aggregation of AGB within this landscape. This index has been previously used to quantify skewness in plant size (Dixon et al. 1987). We have clarified this point in the Methods section (Lines 236-242). 

Dixon, P. M., Weiner, J., Mitchell-Olds, T., & Woodley, R. (1987). Bootstrapping the Gini coefficient of inequality. Ecology, 68(5), 1548-1551.

Line 452: what type of disturbance? Grazing?

R= We specify now as “feeding and activity” and include smaller and larger mammals, Lines 472-475.

Supporting information 

S1 Table species abbreviation – add a column for species life cycle (perennial, annual or biannual)

R = Done.

---

## [Editor Report · Decision Letter 1]

11 May 2021

Hydrological and topographic determinants of biomass and species richness in a Mediterranean-climate shrubland

PONE-D-20-36492R1

Dear Dr. Rodrigo Méndez-Alonzo,

We’re pleased to inform you that your manuscript has been judged scientifically suitable for publication and will be formally accepted for publication once it meets all outstanding technical requirements.

Kind regards,

Wang Li

Academic Editor

PLOS ONE

---

## [Editor Report · Acceptance letter]

18 May 2021

PONE-D-20-36492R1 

Hydrological and topographic determinants of biomass and species richness in a Mediterranean-climate shrubland 

Dear Dr. Méndez-Alonzo:

I'm pleased to inform you that your manuscript has been deemed suitable for publication in PLOS ONE. Congratulations! Your manuscript is now with our production department. 

Kind regards, 

on behalf of

Dr. Wang Li 

Academic Editor

PLOS ONE